# Argmax Centroids: with Applications to Multi-domain Learning

**Chengyue Gong**     **Mao Ye**     **Qiang Liu**
Computer Science Department, The University of Texas at Austin
{cygong17,my21,lqiang}@cs.utexas.edu

## Abstract

We propose a general method to construct centroid approximation for the distribution of maximum points of a random function (a.k.a. argmax distribution), which finds broad applications in machine learning. Our method optimizes a set of centroid points to compactly approximate the argmax distribution with a simple objective function, without explicitly drawing exact samples from the argmax distribution. Theoretically, the argmax centroid method can be shown to minimize a surrogate of Wasserstein distance between the ground-truth argmax distribution and the centroid approximation under proper conditions. We demonstrate the applicability and effectiveness of our method on a variety of real-world multi-task learning applications, including few-shot image classification, personalized dialogue systems and multi-target domain adaptation.

## 1   Introduction

Many problems in machine learning and statistics involve optimizing a random function such as the empirical loss. This work focuses on approximating the distribution of the optimum points of a random function of interest. Specifically, we consider the following problem:

**The Argmax Centroid Problem**   *Assume we are given a random function $f_\xi(\theta)$ where $\xi$ is a random variable and $\theta$ is a variable of interest. We are interested in estimating the distribution $\rho^*$ of the minimum points $\theta_\xi := \arg\min_\theta f_\xi(\theta)$ as we draw $\xi$ randomly. In particular, we want to find a set of "centroid points" $\{\theta_i\}_{i=1}^n$, whose empirical measure is close to $\rho^*$ in terms of Wasserstein distance. We assume the minimizer of $f_\xi$ is always unique.*

This problem can find applications in a variety of machine learning and statistics techniques, including bootstrap (Efron & Tibshirani, 1994), random MAP (Hazan et al., 2013), Thompson sampling (Russo et al., 2017), as well as multi-task learning and meta learning. One naive method to approximate $\rho^*$, which is widely used in the methods above, is to draw an *i.i.d.* sample of $\xi$ and calculate the corresponding argmin points. However, this approach is computationally expensive because calculating each argmin point requires solving an independent optimization problem and the quality of Monte Carlo approximation is poor unless $n$ is very large, especially for large-scale optimization problems in deep learning.

In this work, we propose a more efficient centroid approximation to replace the Monte Carlo sampling, in which we explicitly optimize a set of points $\boldsymbol{\theta} = \{\theta_i\}_{i=1}^n$ to approximate target distribution $\rho^*$. By carefully choosing the location of each point $\theta_i$ (rather than drawing them randomly), our method allows us to obtain a set of points $\boldsymbol{\theta}$ that are well aligned to approximate the overall argmin distribution $\rho^*$. Theoretically, under proper conditions, our method can be viewed as minimizing a surrogate of the Wasserstein distance between the empirical distribution of the centroid approximation and the ground-truth distribution. Therefore, given a small budget on sample size, we can estimate the argmin distribution more accurately than using Monte Carlo estimation.

35th Conference on Neural Information Processing Systems (NeurIPS 2021).

Argmax centroids can find applications in a variety of machine learning problems. In the empirical studies of this work, we mainly focus on meta learning and multi-task learning tasks. For these problems, we learn from a distribution of datasets/tasks/domains and $\xi$ denotes a random dataset drawn from some population. Using argmax centroids allows us to obtain an ensemble of centroid models that can capture the uncertainty and variation in different datasets and domains. We test our method on multiple tasks, e.g., the few-shot image classification, the personalized dialogue system, and the multi-target domain adaptation. For all the above tasks, our method can be easily implemented and enhance the performance of a number of baseline methods, including but not limited to some of the recent state-of-the-art (SOTA) methods, e.g. IFSL (Yue et al., 2020).

## 2 Methodology

We introduce our problem description, propose our algorithm to approximate argmin distributions, and study its properties.

**Problem description**   Many problems in machine learning can be framed as minimizing or maximizing random functions $f_\xi(\theta)$ where $\xi$ is some random variable drawn from a distribution $\pi$ on a space $\Xi$ and $\theta \in \Theta \subseteq \mathbb{R}^d$. That is,

$$\theta_\xi = \arg\min_{\theta \in \Theta} f_\xi(\theta). \tag{1}$$

We assume the minimum of $f_\xi(\theta)$ is unique. Denote by $\rho^*$ the distribution of $\theta_\xi$ where $\xi \sim \pi$, which we call the argmin distribution of $f$. We are interested in approximating $\rho^*$. The problem in (1) finds wide applications in machine learning in various different ways, including random MAP (Hazan et al., 2013), bootstrapping (Efron & Tibshirani, 1994), and multi-domain/meta learning as we elaborate in Section 3.

As mentioned in Section 1, a naive method to approximate $\rho^*$ is to draw an *i.i.d.* sample $\{\xi_i\}_{i=1}^n$ from $\pi$ and calculate the corresponding argmin points $\theta_{\xi_i}$ for each particle respectively. However, the quality of Monte Carlo approximation is poor when $n$ is small (Kuo & Sloan, 2005). In practice, it is not affordable to draw a large number of argmin points for large-scale optimization problems in deep learning, both because that it requires to solve an optimization for calculating each $\theta_{\xi_i}$, and it is memory-hungry to store a large number of $\theta_{\xi_i}$ when the parameter dimension is high.

**Argmax Centroids**   We propose a more efficient centroid approximation to replace Monte Carlo sampling, where we explicitly optimize a set of points $\boldsymbol{\theta} = \{\theta_i\}_{i=1}^n \in \Theta^n$, associated with a set of weights $\boldsymbol{\nu} = \{\nu_i\}_{i=1}^n \in \mathbb{R}^n$, such that the ground-truth target distribution $\rho^*$ can be well approximated by the weighted empirical measure:

$$\hat{\rho}_{\boldsymbol{\theta},\boldsymbol{\nu}} = \sum_{i=1}^n \nu_i \delta_{\theta_i},$$

where $\delta_{\theta_i}$ is delta measure centered at $\theta_i$, and $\boldsymbol{\nu}$ is assumed to take values from the probability simplex on $[n]$: $\mathcal{V} = \{\boldsymbol{\nu}: \sum_{i=1}^n \nu_i = 1, \ \nu_i \geq 0, \ \forall i \in [n]\}$. Here $[n] := \{1, 2, \cdots, n\}$.

Ideally, we would like to choose $\boldsymbol{\theta}, \boldsymbol{\nu}$ to minimize certain distance metrics between $\hat{\rho}_{\boldsymbol{\theta},\boldsymbol{\nu}}$ and $\rho^*$, a canonical example of which is the $p$-Wasserstein distance ($p > 0$), defined as

$$W_p(\hat{\rho}, \rho^*) = \inf_{\mu \in \Pi(\hat{\rho}, \rho^*)} \mathbb{E}_{(\theta, \theta_\xi) \sim \mu} \left[\|\theta - \theta_\xi\|^p\right]^{1/p},$$

where $\Pi(\hat{\rho}, \rho^*)$ denotes the set of probability measures on $\Theta \times \Theta$ such that its two marginal distributions on $\Theta$ are $\hat{\rho}$ and $\rho^*$. We assume that $\|\cdot\|$ is the standard Euclidean norm in this paper.

However, it is expensive to calculate the Wasserstein distance due to the high computational cost of drawing $\theta_\xi \sim \rho^*$, which requires to repetitively solve optimization (1). To improve the computational efficiency, we propose to minimize the following surrogate of Wasserstein distance:

$$\min_{\boldsymbol{\theta} \in \Theta^n, \boldsymbol{\nu} \in \mathcal{V}} W_f(\hat{\rho}_{\boldsymbol{\theta},\boldsymbol{\nu}}, \ \pi), \qquad\qquad W_f(\hat{\rho}, \ \pi) := \inf_{\mu \in \Pi(\hat{\rho}, \pi)} \mathbb{E}_{(\theta, \xi) \sim \mu} \left[f_\xi(\theta)\right], \tag{2}$$

where $\Pi(\hat{\rho}, \pi)$ is the set of probability measures on $\Theta \times \Xi$, whose marginal distributions on $\Theta$ and $\Xi$ equal $\hat{\rho}$ and $\pi$, respectively. Here we replace the norm $\|\theta - \theta_\xi\|^p$ with the function $f_\xi(\theta)$, so that we

do not need to draw $\theta_\xi \sim \rho^*$ and hence solve the expensive optimization in (1). A key property of $W_f$ is that its global minimum in the space of distributions is achieved by the argmin distribution $\rho^*$. Hence, we obtain increasingly accurate estimate of $\rho^*$ as we solve (2) with increasing particle size $n$.

To introduce the result, we start with a general definition of argmin distributions that works for when the minimum of $f_\xi$ is not necessarily unique.

**Definition 2.1.** *Assume $f_\xi^* := \min_{\theta \in \Theta} f_\xi(\theta) > -\infty$ for every $\xi \in \Xi$. We say that a distribution $\rho^*$ on $\Theta$ is an argmin distribution of $f_\xi$ with $\xi \sim \pi$, if there exist a coupling measure $\mu^* \in \Pi(\rho^*, \pi)$, such that $\rho^*(\theta \in \arg\min f_\xi) = 1$. Here $\arg\min f_\xi$ is the set of global minima of $f_\xi$, that is, $\arg\min f_\xi = \{\vartheta \in \Theta : f_\xi(\vartheta) = f_\xi^*\}$.*

If the optimum of $f_\xi$ is unique for every $\xi \in \Xi$, then there is an unique argmin distribution $\rho^*$, which is the distribution of $\theta_\xi = \arg\min_{\theta \in \Theta} f_\xi(\theta)$ when $\xi \sim \pi$.

**Theorem 2.2.** *Assume $\rho^*$ is any argmin distribution of $f_\xi$ with $\xi \sim \pi$. Then the minimum of $W_f(\rho, \pi)$ is achieved by $\rho^*$, that is,*

$$W_f(\rho^*, \pi) = \mathbb{E}_{\xi \sim \pi}\left[\min_{\theta \in \Theta} f_\xi(\theta)\right] = \min_{\rho \in \mathcal{P}} W_f(\rho, \pi),$$

*where $\mathcal{P}$ is the space of probability distributions on $\Theta$.*

It is not easy to directly minimize the form of $W_f$ in (2) because it requires to jointly minimize $\boldsymbol{\theta}, \boldsymbol{\nu}$ and the coupling measure $\mu$. However, the key observation below shows that the optimization in both $\boldsymbol{\nu}$ and $\mu$ can be solved in closed form, yielding a simple objective function on $\boldsymbol{\theta}$.

**Theorem 2.3.** *For a fixed $\boldsymbol{\theta} \in \Theta^n$, we have*

$$\min_{\boldsymbol{\nu} \in \mathcal{V}} W_f(\hat{\rho}_{\boldsymbol{\theta}, \boldsymbol{\nu}}, \pi) = \mathbb{E}_{\xi \sim \pi}\left[\min_{i \in [n]} f_\xi(\theta_i)\right],$$

*where the minimum on the left hand side is achieved by $\boldsymbol{\nu}^* = \{\nu_i\}_{i=1}^n$ with*

$$\nu_i^* = \mathbb{E}_{\xi \sim \pi}\left[\mathbb{P}(i \in \arg\min_{j \in [n]} f_\xi(\theta_j))\right], \quad \forall i \in [n], \tag{3}$$

*with $\mathbb{P}(i \in \arg\min_{j \in [n]} f_\xi(\theta_j)) := \frac{1}{Z_{\boldsymbol{\theta}, \xi}} \mathbb{I}(i \in \arg\min_{j \in [n]} f_\xi(\theta_j))$ and $Z_{\boldsymbol{\theta}, \xi}$ the number of elements in the argmin set, i.e., $Z_{\boldsymbol{\theta}, \xi} = |\arg\min_{j \in [n]} f_\xi(\theta_j)|$.*

Here the $\nu_i^*$ in (3) is the probability that $\theta_i^*$ achieves the minimum value of $\min_{j \in [n]} f_\xi(\theta_j^*)$, with ties broken randomly with equal probabilities.

Therefore, the optimal centroids $\boldsymbol{\theta}^*$ and the weights $\boldsymbol{\nu}^*$ are given by

$$\boldsymbol{\theta}^* = \arg\min_{\{\theta_i\}_{i=1}^n}\left\{L(\boldsymbol{\theta}) := \mathbb{E}_{\xi \sim \pi}\left[\min_{i \in [n]} f_\xi(\theta_i)\right]\right\}, \quad \nu_i^* = \mathbb{E}_{\xi \sim \pi}\left[\mathbb{P}(i \in \arg\min_{j \in [n]} f_\xi(\theta_j^*))\right]. \tag{4}$$

Note that objective in (4) reduces to the well known K-means objective function if we take $f_\xi(\theta) = \|\theta - \theta_\xi\|^2$, in which case $\boldsymbol{\theta}$ and $\boldsymbol{\nu}$ are the centers and the sizes of the clusters, respectively. Intuitively, minimizing $L(\boldsymbol{\theta})$ allows us to ensure that the best function value $\min_{i \in [n]} f_\xi(\theta_i)$ is small on average.

In practice, we can optimize $\theta_i$ using stochastic gradient descent, and update $\nu_i$ recursively. See Algorithm 1 for details. At each iteration, we draw a (set of) $\xi \sim \pi$, find the $\theta_i$ that attains $i = \arg\min_j f_\xi(\theta_j)$ and update $\theta_i$ with gradient descent as displayed in line 4, Algorithm 1. The frequency $\nu_i$ is also updated accordingly when a $\theta_i$ is updated, see line 5 in Algorithm 1.

**Re-sampling** During optimization, it is possible that some $\theta_i$ are rarely selected and hence its weight $\nu_i$ becomes small and causes a degeneration problem (i.e., nearly zero important weight). To address this problem, we monitor the effective sample size of the points, defined as $n_{eff} = (\sum_i \nu_i)^2 / (\sum_i \nu_i^2)$. If the effective sample size becomes smaller than a threshold, we re-sample a set of new particles $\theta_i$ with replacement from $\hat{\rho}_{\boldsymbol{\theta}, \boldsymbol{\nu}} = \sum_{i=1}^n \nu_i \delta_{\theta_i}$ and repeat the updates. In practice, we break the ties randomly in argmin to avoid the case when two $\theta_i$ remain to be identical throughout the algorithm.

---
**Algorithm 1** Main Algorithm: Argmax Centroids for Approximating $\rho^*$
---
1: **Input**: $\{\theta_i\}_{i=1}^n$ denotes the learnable centroids, $\nu_i$ denotes the updated frequency of $\theta_i$, $\epsilon$ denotes the step size for gradient descent, $\alpha \in [0, 1]$ denotes a parameter to control the frequency update and $\eta$ is a threshold to control the resampling.
2: **while** not convergence **do**
3:     Find $i_* = \arg\min_i f_\xi(\theta_i)$ with $\xi \sim \pi$.
4:     Update $\theta_{i_*} \leftarrow \theta_{i_*} - \epsilon \nabla_{\theta_{i_*}} f_\xi(\theta_{i_*})$, $\nu_{i_*} \leftarrow \alpha\nu_{i_*} + 1$, and $\nu_i = \alpha\nu_i$ for $i \neq i_*$.
5:     If $n_{eff} \leq \eta$, resample $\{\theta_i\}_{i=1}^n$ from $\hat{\rho}_{\boldsymbol{\theta},\boldsymbol{\nu}} \propto \sum_{i=1}^n \nu_i \delta_{\theta_i}$.
6: **end while**
---

**Bounds with Wasserstein**   Because $W_f$ coincides with the Wasserstein distance if $f_\xi$ is a simple quadratic function $f_\xi(\theta) = ||\theta - \theta_\xi||^2$, we can provide some simple bounds between $W_f$ and Wasserstein distance by approximating $f_\xi$ with the quadratics using Taylor approximation.

**Assumption 2.4.** *Let $\theta_\xi$ be the minimum of $f_\xi(\theta)$ as in* (1)*. Assume there exists $h_1, h_2, p_1, p_2 \in (0, \infty)$, such that for any $\theta \in \Theta$ and $\xi \in \Xi$,*

$$h_1 \|\theta - \theta_\xi\|^{p_1} \leq f_\xi(\theta) - f_\xi(\theta_\xi) \leq h_2 \|\theta - \theta_\xi\|^{p_2}.$$

This assumption holds with $p_1 = p_2 = 2$ if $f$ is strongly convex w.r.t. $\theta$ and $0 < h_1/2 \leq \lambda_{min}(f''_\xi(\theta)) \leq \lambda_{max}(f''_\xi(\theta)) \leq h_2/2 < \infty$ for $\forall\theta$ and $\xi$, where $\lambda_{min}$ and $\lambda_{max}$ are the minimum and maximum eigenvalue; it also holds with $p_2 = 1$ if $f$ is Lipschitz w.r.t. $\theta$ with $\|f\|_{Lip} = h_2$.

**Theorem 2.5.** *Assume Assumption 2.4 holds.*

*1) We have for any $\boldsymbol{\theta} \in \Theta^n$ and $\boldsymbol{\nu} \in \mathcal{V}$,*

$$h_1 W_{p_1}(\hat{\rho}_{\boldsymbol{\theta},\boldsymbol{\nu}}, \rho^*)^{p_1} \leq W_f(\hat{\rho}, \pi) - W_f(\rho^*, \pi) \leq h_2 W_{p_2}(\rho_{\boldsymbol{\theta},\boldsymbol{\nu}}, \rho^*)^{p_2}.$$

*2) For the optimal $\boldsymbol{\theta}^*$ and $\boldsymbol{\nu}^*$ in* (4)*, we have*

$$W_{p_1}(\hat{\rho}_{\boldsymbol{\theta}^*,\boldsymbol{\nu}^*}, \rho^*)^{p_1} \leq \frac{h_2}{h_1} \inf_{\boldsymbol{\theta} \in \Theta^n, \boldsymbol{\nu} \in \mathcal{V}} W_{p_2}(\rho_{\boldsymbol{\theta},\boldsymbol{\nu}}, \rho^*)^{p_2}. \tag{5}$$

## 3   Applications and Related Works

Approximating argmax distribution can be found as a key component of many machine learning methods. This includes methods like bootstrap (Ye & Liu, 2021), Thompson sampling, and random MAP, in which the argmax distribution plays a key role by design, and as well as novel applications from meta learning and multi-task learning, which we study extensively in Section 4.

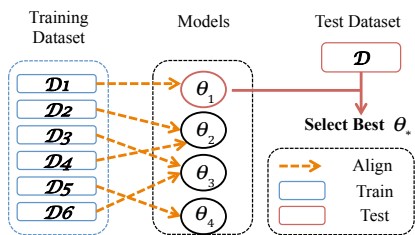

Figure 1: An illustration of our selection process for both training and evaluation. Test dataset $\mathcal{D}$ refers to the training or validation subset of the coming test task $\mathcal{D}$.

**Learning with Ensembles of Datasets**   Approximating the argmax distribution finds a natural application in meta learning and multi-task/multi-domain learning, in which we need to learn from an ensemble or a distribution of datasets. Denote by $D$ a dataset on which we want to learn a model parameter $\theta$ with a loss function $\ell(\theta, D)$. Instead of having a single dataset, we consider the case when we have a large number of datasets $\{D_i\}$ which we assume is drawn from a distribution $\mathcal{P}$.

We aim to find a set of models $\{\theta_i\}_{i=1}^n$ where exists at least one good model for every dataset $D$ with small $n$. Denote by $\theta_D = \arg\min_\theta \ell(\theta, D)$ and $\rho^*$ the distribution of $\theta_D$ as $D \sim \mathcal{P}$. Then the problem can framed as finding $\{\theta_i\}_{i=1}^n$ to approximate $\rho^*$, and hence can be framed into

$$\min_{\{\theta_i\}_{i=1}^n} \left\{ \mathbb{E}_{D \sim \mathcal{P}} \left[ \min_{i \in [n]} \ell(\theta_i, D) \right] \right\}. \tag{6}$$

In the test phase, given a new coming dataset $D$, which we assume includes a validation set $\tilde{D}$ with true labels, we select the best model $\theta_{i_*}$ among the centroids that minimizes the loss $i_* = \arg\min_{i \in [n]} \ell(\theta_i, \tilde{D})$. The pipeline is shown in Figure 1. The idea here is to prepare a pool of candidate models to best cover all different scenarios (according to probability $\theta_D \sim \rho^*$), so that we can select the best model for any new random task drawn from $\rho^*$. This is different from typical ensemble learning which averages the output of multiple models during evaluation; instead, we use the best single model $\theta_{i_*}$ selected from the pool during evaluation. The objective $\ell$ and how to optimize $\ell$ is problem-dependent. For example, in meta learning, $\ell$ is updated with gradient descent and meta gradient. The details about the formulation of $\ell$ and its update rule is shown in Section 4 for different tasks. The method in (6) can also be used in bootstrapping, in which we are given a dataset $D_0$ of true interest and we draw random datasets $D \sim \mathcal{P}$ by resampling from $D_0$ with replacement. Using the argmax centroids can yield better approximation with small $n$ than naive random sampling that calculates $\theta_D$ from randomly drawn $D \sim \mathcal{P}$. The theoretical properties and efficient algorithms specified to bootstrap desires an independent treatment, which we explore separately in another work. Another related method is multiple choice learning (MCL) (Guzman-Rivera et al., 2012; Yu et al., 2018; Lee et al., 2016), which is an ensemble learning method using a similar objective function as (6) but with very different motivation and settings. MCL can be viewed as minimizing (6), but assuming each $D$ to be a single data example rather than a dataset, with the goal of learning a set of models and ensemble them for a single dataset. During evaluation, the outputs of the multiple models from MCL are averaged to give the final output, rather than selecting the best one as we suggest above based on the perspective of approximating $\rho^*$.

**Multi-task Learning**    Jointly learning models from multiple tasks has attracted long-term attention in machine learning community. In multi-task learning (MTL), there are multiple learning tasks and most of them (or all of them) are assumed to be related to each other. MTL algorithms can be grouped into two main approaches: the feature learning methods and task relation learning methods. Since tasks are related, it is intuitive to assume that different tasks share some common feature representation, and thus feature learning approaches focus on learning the common representation with regularizations (Yang et al., 2009), structure designing (Caruana, 1997), etc. The other approach focuses on learning task relations. The task relations are learnt from data automatically with a given prior (Yu et al., 2005; Xue et al., 2007; Zhang & Yeung, 2012), matrix decomposition (Chen et al., 2012) or clustering (Thrun & O'Sullivan, 1996). Recently, meta learning (Finn et al., 2017), lifelong learning (Thrun, 1998) and continual learning (Zenke et al., 2017) are proposed to solve MTL under different conditions.

Our method is similar to learning task relation with clustering. Different from previous works which cluster the tasks with latent representation (Zhou & Zhao, 2015), learnable relation matrix (Zhou et al., 2011; Kang et al., 2011) or side information (Caruana, 1997), we automatically assign a model to a certain domain/task based on its loss.

# 4  Experiments

In our experiments, we start with a toy example on which we show that our algorithm approximates the argmax distribution better than simple Monte Carlo estimation (Section 4.1). We then apply our method to standard few-shot learning benchmarks, in which parameters are learned on thousands of domains (Section 4.2). We further apply our method to personalized dialogue systems (Peng et al., 2020) (Section 4.3) and multiple target domain adaptation (Yu et al., 2018) (Section 4.4) to verify whether our proposed method can be applied to different real-world applications. In all experiments, we set the replacement controller $\eta = 1.2$ and $\alpha = 0.5$ for Algorithm 1.

## 4.1  Toy Examples

We start with a toy example to verify that our method can provide a better approximation of $\rho^*$ than Monte Carlo estimation with the same number of samples. We set $f_\xi(\theta) = \|\theta - \xi\|^2$, so that $\theta_\xi = \xi$ and $\rho^* = \pi$. We set $\pi$ to be $d$-dimension Gaussian mixture model with five modes. Note that our method is equivalent to a gradient descent variant of K-means in this case. Figure 2 demonstrates the two-dimension case, from which we can see that argmax centroids captures the different modes in the distribution and are aligned optimally to get better approximation then random sampling.

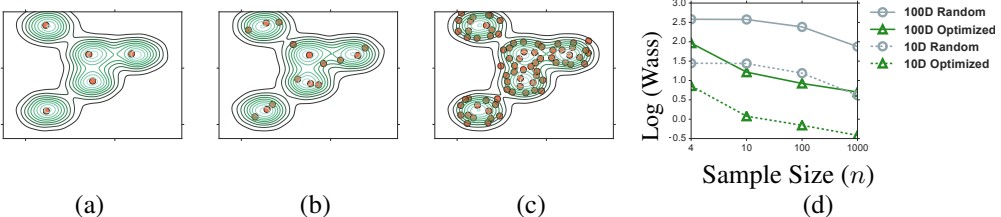

(a)      (b)      (c)      (d)

Figure 2: 2D toy example. Figure (a), (b) and (c) show the estimated centroids $\{\theta_i\}_{i=1}^n$ (the red dots) and contour (green) of the true distribution $\rho^*$. Figure(d) shows the 2-Wasserstein distance ($p = 2$) between the empirical measure $\hat{\rho}_{\theta^*, \nu^*}$ of the centroids (centroid approximation) and the true distribution $\rho^*$ w.r.t. the sample size $n$, when we vary the dimension in $d \in \{10, 100\}$. 'Log (Wass)' denotes log-scale Wasserstein distance.

Figure 2(d) shows the 2-Wasserstein distance ($p = 2$) from the empirical measure $\hat{\rho}_{\theta^*, \nu^*}$ of the centroids and the true distribution $\rho^*$ (calculated by drawing a large sample from $\rho^*$), when the dimension $d$ of $\theta$ varies. We can see that for both 100-dimension and 10-dimension cases, our method yields lower Wasserstein distance than random sampling for different $n$.

| Method | Transductive | Backbone | 1-shot (Acc) | 5-shot (Acc) |
|---|:---:|:---:|:---:|:---:|
| MAML (Finn et al., 2017) | ✓ | Conv4 | 48.7±1.8 | 64.6±1.2 |
| (Tian et al., 2020) | ✓ | ResNet12 | 64.8±0.6 | 82.1±0.4 |
| Baseline (Dhillon et al., 2020) | × | WRN-28-10 | 68.1±0.7 | 80.4±0.5 |
| SIB (Hu et al., 2020) | ✓ | WRN-28-10 | 70.0±0.6 | 80.0±0.3 |
| + Ours | ✓ | WRN-28-10 | **71.2±0.6** | **81.1±0.4** |
| IFSL (Yue et al., 2020) | ✓ | WRN-28-10 | 73.2±0.3 | 82.9±0.4 |
| + Ours | ✓ | WRN-28-10 | **73.6±0.4** | **83.3±0.3** |

Table 1: Few-shot learning results averaged over 3 trials on Mini-ImageNet.

| Method | Transductive | Backbone | 1-shot (Acc) | 5-shot (Acc) |
|---|:---:|:---:|:---:|:---:|
| MAML (Finn et al., 2017) | ✓ | Conv4 | - | - |
| (Tian et al., 2020) | ✓ | ResNet12 | 71.5±0.2 | 86.0±0.5 |
| Baseline (Dhillon et al., 2020) | × | WRN-28-10 | 72.9±0.1 | 86.2±0.5 |
| SIB (Hu et al., 2020) | ✓ | WRN-28-10 | 81.2±0.6 | 87.1±0.4 |
| + Ours | ✓ | WRN-28-10 | **81.8±0.6** | **87.6±0.4** |
| IFSL (Yue et al., 2020) | ✓ | WRN-28-10 | 82.4±0.5 | 88.3±0.5 |
| + Ours | ✓ | WRN-28-10 | **82.7±0.5** | **88.5±0.5** |

Table 2: Few-shot Learning results averaged over 3 trials on TieredImageNet.

## 4.2 Few-Shot Supervised Learning

We now apply our method to few-shot image classification using the formulation in (6), in combination with model agnostic meta learning (MAML) (Finn et al., 2017).

To introduce the setting, let $D = \{D_j^{Train}, D_j^{test}\}$ be a random dataset (or task) drawn $\mathcal{P}$, which includes a training and test set. Let $\theta' = \mathcal{A}(\theta, D^{train})$ be the result of training the parameter on $D^{train}$ starting from $\theta$, where $\mathcal{A}$ denotes the training algorithm, which is typically one step or few steps of stochastic gradient descent with the training set. In MAML, we optimize $\theta$ so that the peformance of $\theta' = \mathcal{A}(\theta, D^{train})$ is maximized on $D^{test}$. The MAML loss is $\ell(\theta, D) = L(\mathcal{A}(\theta, D^{train}), D^{test})$, with the hope to rapidly solve a new coming task after learning several other similar tasks.

Standard MAML learns a single model $\theta$, which may not work well for all the tasks $D$ when the discrepancy between tasks is large. By plugging the MAML loss into (6), we obtain a new generalization of MAML with which we can prepare a pool of candidate models $\{\theta_i\}_{i=1}^n$ (the argmax centroids), out of which we can select the best one during testing. Note that this is different from the

existing ensemble MAML methods such as (e.g., Lee et al., 2019; Yoon et al., 2018), which averages (instead of selecting the best among) the outputs of multiple models.

**Baselines** We compare and implement our method based on the recent state-of-the-art few-shot learning methods (e.g Hu et al., 2020; Yue et al., 2020; Wang et al., 2019). These methods train a network consisting of a backbone network shared by all the datasets (tasks), which is fed into classification heads specialized to individual datasets. During training, we first train a backbone network on *all* the training datasets in the training tasks using typical method, and then use meta-learning style training to tune the classification heads while keeping the backbone frozen.

We implement our proposed argmax centroids upon two recently proposed few-learning methods, SIB and IFSL. These recently proposed methods achieve state-of-the-art results on supervised few-shot image classification benchmarks [1]. SIB (Hu et al., 2020) proposes to learn synthetic gradient for new coming tasks, and IFSL (Yue et al., 2020) finds out that pre-trained knowledge is essentially a confounder that causes spurious correlations between the sample features and class labels in support set. They further propose regularization terms to eliminate the mismatch.

| Method | MAML (Finn et al., 2017) | Ensemble (Yoon et al. 2018) | Bayesian (Yoon et al. 2018) | Ours |
|---|---|---|---|---|
| ResNet-18 | 56.6±.3 | 56.7±.3 | 57.0±.4 | **57.4±.2** |
| WRN-28-10 | 57.4±.4 | 57.4±.5 | 57.7±.4 | **58.1±.4** |

Table 3: Comparison to ensemble MAML methods on Mini-ImageNet. We demonstrate the accuracy for one-shot classification with ResNet-10 and WRN-28-10. We set the number of heads to 16 while sharing the backbone parameters. 'Ensemble' and 'Bayesian' denotes ensemble MAML and Bayesian MAML, respectively.

All these baselines and other recent works (e.g. Hu et al., 2020; Yue et al., 2020; Wang et al., 2019; Dhillon et al., 2020) focus on learning a better single model by introducing strong regularization methods. For the baselines, we focus on, a clear disadvantage of SIB and IFSL is that they share one single $\theta$ for all the datasets and may not work well once the discrepancy between task is large. By applying our method, we train a fixed number of classification heads, which are treated as the argmax centroids. This allows us to pick the best head $\theta$ and further finetune it on the training set for each test task during testing. We simply combine these baseline approaches with our algorithm to obtain further improvements. To save computation cost in practice, we evaluate $\mathcal{A}(\theta, D^{Train})$ by one-step gradient descent and then select $\theta_i$ with lowest loss among $\boldsymbol{\theta}$.

**Datasets** Standard benchmarks of few-shot classification are chosen for experiments. We evaluate all the baselines and our algorithms on two subsets of ImageNet, Mini-ImageNet and TieredImageNet (Sun et al., 2019). Mini-ImageNet contains 64 classes for training, 16 for validation and 20 for test. Training/validation/test tasks are sampled from the 64/16/20 classes. TieredImageNet (Sun et al., 2019) is much larger compared to miniImageNet and other benchmarks, with 608 classes and 1300 samples per class. These classes are partitioned into 351/97/160 disjoint sets for train/val/test to achieve a larger domain difference between training and testing During train-

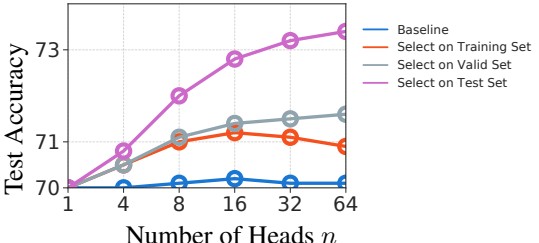

Figure 3: Training $n$ different heads for the few-shot learning task MiniImageNet with SIB (Hu et al., 2020). 'Baseline' denotes training one model , creating $\{\theta_i\}_{i=1}^n$ by dropout and selecting $\{\theta_i\}_{i=1}^n$ during evaluation.

ing, the images in Mini-ImageNet and tieredImageNet are resized to 80×80 and 84×84, respectively. Each sampled dataset $D$ is a 5-class classification, with 1-shot/5-shot images per class for training and 15 images per class for test. When evaluating the trained models, 2000 test tasks are sampled from the test set, and the accuracy on $D^{test}$ of all the test tasks are reported for comparison.

---

[1] https://few-shot.yyliu.net

**Implementation Details**   For each baseline method, we create a small number of additional parameters. During the second training stage, given a backbone, e.g., Wide-ResNet-28-10 (Zagoruyko & Komodakis, 2016), we use additional parameters $\{\theta_i\}_i^n$ to re-weight the final-layer features. In the experiments, for few-shot learning based on SIB and IFSL, we set $n = 16$. We only introduce a small number of additional parameters (0.01M parameters) compared to the model size (22M parameters).

For both the training and testing, we keep all the settings (e.g. optimizer, learning rate, number of tasks) the same for each baseline method. We first train the feature extractors using the standard cross-entropy loss on the base classes (i.e., the classes in the training set in all training tasks). Secondly, training the heads following the standard meta learning pipeline. Finally, we test the trained models on sampled test tasks. For each test task, we do a few gradient descent steps on the training set. The number of iterations follows the baselines.

**Results**   Table 3 shows the comparison between our proposed method and standard ensemble MAML methods. We implement our method upon several recently proposed strong baselines. As demonstrated in Table 1 and 2, our algorithm consistently improves all the baseline methods in all settings, which suggests that our proposed algorithm is a general method for few-shot image classification. For example, on Mini-ImageNet, we improve the accuracy of SIB by a large margin for both 1-shot (70.0% to 71.2%) and 5-shot (80.0% to 81.1%) classification. For IFSL, we can also boost the performance under most of the settings.

**Comparison to Ensemble Methods**   Here, we follow the setting in and ensemble MAML (Yoon et al., 2018) where the backbone is not pretrained on all the training images and is jointly training during the meta learning step. We compare two popular ensemble baselines, ensemble MAML and Bayesian MAML (Yoon et al., 2018). Ensemble MAML trains $n$ models independently and take an average of the outputs of all the models during evaluation. Bayesian MAML (Yoon et al., 2018) uses SVGD (Liu & Wang, 2016) to jointly update $n$ models during training and ensemble them during evaluation. Instead of choosing the best model for a test domain like our method, the ensemble methods average the outputs of all the models. From Table 3, we can see that our method is better than both ensemble approaches on one-shot classification on Mini-ImageNet with different neural architectures. On ResNet-18 and WRN-28-10, our method achieves better results than ensemble methods.

**Ablation Study**   In Figure 3, we change $n$ and make a selection based on training/test/validation loss to have a deeper understanding of our algorithm. We notice that, if we do not have a separate validation set, we may need to select the best model on the training set during evaluation, which cause overfitting. We investigate this issue in Figure 3, in which we show the results when we select the best models using training, validation and, test set, respectively. In order to do the selection on the validation set, we additionally draw 5 images per class for each test task. The result shows that, in our case, both selecting on training and validation set is worse than the "oracle" way (i.e., selecting on the test set and evaluating on the test set). When the number of heads increases, the performance of selecting on training set drops and over-fitting happens. However, in practice, one cannot get access to the test set during training. Therefore, selecting on the validation set is an alternative choice.

## 4.3  Personalized Dialogue System

**Problem Set**   We consider learning personalized chit-chat dialogue agents for making chat-bots more consistent for each user (e.g. Zhang et al., 2018; Madotto et al., 2019b; Tian et al., 2020). Previous approaches usually learning persona similarity from persona descriptions (Zhang et al., 2018). Different from these methods, current personalized dialogue systems focus on learning to quickly adapt to new personas by using few samples. In this problem, each dataset $D = (D^{Train}, D^{Test}) \sim \mathcal{P}$ contains several dialogues of a user. A dialogue contains a set of utterances, and the objective is to predict the next utterance given previous utterances. We use the MAML objective for training.

| Method | PPL ↓ | BLEU ↑ |
|---|---|---|
| PAML (Madotto et al., 2019a) | 46.35 | 0.77 |
| Ours | **43.94** | **0.91** |

Table 4: Results of automatic evaluation on personalized dialogue generation. We report case-insensitive BLEU and PPL.

**Baselines**  We use a recently-proposed meta learning dialogue system (Madotto et al., 2019b) as the baseline. We measure the difference between the ground-truth output and model output to evaluate the model performance by PPL (Perplexity) and case-insensitive BLEU score (Papineni et al., 2002).

**Implementation Details**  Before training, the datasets sampled from $\mathcal{P}$ is split into meta-training, meta-validation and meta-test dataset. The meta-learner learns how to learn by training and evaluating on the meta-training set. Meta-learning hyper-parameters are tuned on the meta-validation set. The meta-test set measures generalization on new, unseen tasks. We use the BERT-base (Devlin et al., 2019) model as the backbone and tune the parameters in the backbone together with the final softmax layer. For meta training, we use Adam (Kingma & Ba, 2014) with learning rate $10^{-3}$, $10^{-2}$ for inner and outer loop training, respectively. During the evaluation, for all the models, we used beam search with beam size 4 and length penalty 1.2. These hyper-parameters are selected based on the performance of the meta-validation dataset.

For our method, we create $n = 16$ softmax head (i.e. a linear transform followed by a softmax layer), share the backbone parameters and select one head for each persona with the lowest loss on the test set for this persona during training, and select the softmax head with the lowest loss on the training set during evaluation.

**Result**  Table 4 shows that our proposed algorithm beats the meta-learning baseline on both the PPL and BLEU measure. We improve the BLEU from 0.77 to 0.91 and PPL from 46.35 to 43.94, respectively.

### 4.4  Multi-Target Domain Adaptation

Domain adaptation is a powerful approach for learning under distributional shift between training and test data (e.g. Wu et al., 2018; Tzeng et al., 2017). Here, we show that our proposed algorithm can also be applied to multi-target domain adaptation (Yu et al., 2018) using the dataset proposed in Peng et al. (2019). For multi-target domain adaptation, one source domain and $m$ multi-target domain is given. In this experiment, we use the classical domain adaptation method proposed by Tzeng et al. (2017), in which an adversarial net is used to learn the adaptation from the latent space in the source domain.

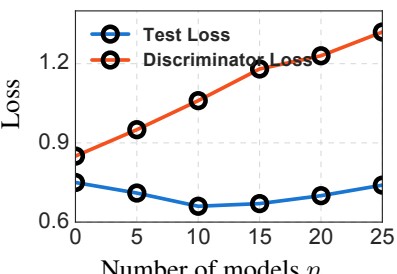

Figure 4:  Illustration of the per-domain test loss, and per domain negative discriminator loss. For the multi-domain adaptation task, when the number $n$ of models increases, the negative discriminator loss always increases while the test loss first decreases and then increases.

We first pretrain a CNN using labeled image examples from the source domain and denote the backbone as $g$. Next, for the target domain, we perform adversarial adaptation by learning a map $\phi$ after $g$ such that a discriminator that sees source examples $g(\cdot)$ and target examples $\phi(g(\cdot))$ cannot reliably predict their domain label. During the evaluation, target images are mapped with backbone $g$ followed by the learned map to the shared feature space and classified by the final-layer classifier trained on the source domain.

**Our Method**  After pretraining a backbone on the source domain, We consider to use $n$ adversarial nets $\{\theta_i\}_{i=1}^n$ as domain adaptors to match the representation of the source domain and target domains. Each $\theta_i = (\phi_i, \gamma_i)$ contains two part of parameters where $\phi_i$ is a feature map and $\gamma_i$ is a discriminator. $\phi_i$ is an MLP whose input and output have the same dimension and $\gamma_i$ maps the input to a one-dimension scalar followed by a sigmoid function.

Denote $z = g(x)$ the features extracted by the backbone and $S$ the source domain, $\ell(\theta_i, D)$ equals to

$$-\mathbb{E}_{z=g(x),x\sim D}\left\{\log\gamma_i(\phi_i(z))+\log(1-\gamma_i(\phi_i(z)))\right\}-\mathbb{E}_{z=g(x),x\sim S}\left\{\log\gamma_i(z)+\log(1-\gamma_i(z))\right\},$$
(7)

is an adversarial loss which forces the discriminator to make uniform prediction.

**Baselines**  We compare our method with three simple baselines. One is using a shared adversarial domain adaptors for all the target domains. Another is using $n$ different domain adaptors for $m$ target domains. The third baseline is randomly assigning domains with models for each iteration during training and average all the outputs for all models during evaluation.

**Experiment Settings**  We use the DomainNet dataset proposed by Peng et al. (2019). DomainNet contains 6 styles, 345 classes and 569,010 images. We create a subset of DomainNet to do experiments. We sample 25 classes in the same style and sample 300 images for each class. For each sample subset (domain), we use 5 different data shifts (e.g. brightness, fog, snow, MotionBlur, Gaussian noise) (Hendrycks et al., 2019) to create new domains. Finally, we obtain 30 different domains. We randomly select one domain data as the source domain and the rest as target domains during experiments. For the neural architecture, we use a standard ResNet-18 (He et al., 2016) to train on the source domain, a three-layer MLP as $\gamma_i$ and another three-layer MLP as $\phi_i$. During training and testing, we resize the image into $64 \times 64$ size. During training, we use standard data augmentation (mirror and flip).

| Method | 1 Adaptor | $m$ Adaptor | Ensemble | Ours |
|--------|-----------|-------------|----------|------|
| Acc ↑ | 39.5±0.4 | 41.8±0.6 | 41.3±0.7 | **44.3±0.6** |

Table 5: Results on multi-target domain adaptation. The reported accuracy are averaged over 5 trials. "Ensemble" denotes randomly assigning models with domains during training and ensembling all the models during evaluation.

**Results**  As demonstrated in Table 5, compared to sharing one domain adaptor for all the domains, using a domain-specific domain adaptor improves the accuracy from 39.5% to 41.8%. With our proposed method, we can further improve the 41.8% accuracy to 44.3%. In Table 5, we also demonstrate that randomly aligning different domains hurts the performance. It indicates that in this setting our method is much better than ensemble multiple models. We further set up an ablation study on the number of heads. As shown in Figure 4, similar to the results in Figure 3, increasing the number of heads can always improve the training loss (i.e., discriminator loss in (7)). Here, increasing $n$ makes the negative discriminator loss higher indicates that it mixes the two domains successfully. However, the test loss increases once $n$ is too large. It demonstrates that our argmax centroids method implicitly serves as a regularization by sharing the same linear map $\phi$ for multiple similar domains. Sharing the same linear map increases the number of training data for each $\theta$, and therefore avoids overfitting.

## 5   Conclusion

In this work, we propose a method to estimate the argmax distribution. It can be an alternative for related methods, and we further apply it to many multi-domain and multi-task learning tasks. Aligning similar domains into a same model, the proposed algorithm boost the SOTA performance on several few-shot classification and meta learning tasks. In the concurrent work (Ye & Liu, 2021) we use the centroid learning idea to improve the bootstrap for data uncertainty quantification, in which we establish a rigorous theoretically result showing that the centroid objective function is uniformly close to Wasserstein distance through the whole optimization trajectory induced by a modified gradient descent algorithm. For the next step, we will consider the connection with diversifying models and other topics. We will also apply our method to more meaningful and challenging tasks, e.g. lifelong learning, continual learning.

**Acknowledgements** The work is conducted in the statistical learning and AI group in computer science department at UT Austin, which is supported in part by CAREER-1846421, SenSE-2037267, EAGER-2041327, and Office of Navy Research, and NSF AI Institute for Foundations of Machine Learning (IFML). We would like to thank the anonymous reviewers for their thoughtful comments and efforts towards improving our manuscript.

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
