# A  Proofs

*Proof of Theorem 2.1.* For each $\mu \in \Pi(\hat{\rho}_{\boldsymbol{\theta},\boldsymbol{\nu}}, \pi)$, define $\mu(\theta = \theta_i \mid \xi) = \nu_{i|\xi}$. Then we have $\{\nu_{i|\xi}\}_{i=1}^n \in \mathcal{V}$ for each fixed $\xi$, and $\nu_i = \mathbb{E}_{\xi \sim \pi}[\nu_{i|\xi}], \forall i \in [n]$. We have

$$\mathbb{E}_{(\theta,\xi)\sim\mu}[f_\xi(\theta)] = \mathbb{E}_{\xi\sim\pi}\left[\sum_{i=1}^n \nu_{i|\xi} f_\xi(\theta_i)\right] \geq \mathbb{E}_{\xi\sim\pi}\left[\min_{i\in[n]} f_\xi(\theta_i)\right].$$

Taking inf on $\mu$ and $\boldsymbol{\nu}$ yields that

$$\inf_{\boldsymbol{\nu}\in\mathcal{V}} W_f(\hat{\rho}_{\boldsymbol{\theta},\boldsymbol{\nu}}, \pi) \geq \mathbb{E}_{\xi\sim\pi}\left[\min_{i\in[n]} f_\xi(\theta_i)\right].$$

On the other hand, for $\nu_i^* = \mathbb{E}_{\xi\sim\pi}\left[\mathbb{P}(i \in \arg\min_{j\in[n]} f_\xi(\theta_j))\right]$, we define a coupling $\mu_{\boldsymbol{\theta},\pi}^*$ such that 1) its marginal on $\mathcal{V}$ equals $\pi$, and 2)

$$\mu_{\boldsymbol{\theta},\pi}^*(\theta = \theta_i \mid \xi) = \mathbb{P}(i \in \arg\min_{j\in[n]} f_\xi(\theta_j)) := \nu_{i|\xi}^*.$$

It is easy to show that $\mu_{\boldsymbol{\theta},\pi}^*$ matches with $\nu_i^*$ in that $\nu_i^* = \mu_{\boldsymbol{\theta},\pi}^*(\theta = \theta_i)$, and hence we have $\mu_{\boldsymbol{\theta},\pi}^* = \Pi(\hat{\rho}_{\boldsymbol{\theta},\boldsymbol{\nu}^*}, \pi)$. With this, we have

$$W_f(\hat{\rho}_{\boldsymbol{\theta},\boldsymbol{\nu}^*}, \pi) \leq \mathbb{E}_{(\theta,\xi)\sim\mu_{\boldsymbol{\theta},\pi}^*}[f_\xi(\theta)]$$

$$= \mathbb{E}_{\xi\sim\pi}\left[\sum_{i=1}^n \nu_{i|\xi}^* f_\xi(\theta_i)\right]$$

$$= \mathbb{E}_{\xi\sim\pi}\left[\min_{i\in[n]} f_\xi(\theta_i)\right].$$

This proves that $\inf_{\boldsymbol{\nu}\in\mathcal{V}} W_f(\hat{\rho}_{\boldsymbol{\theta},\boldsymbol{\nu}}, \pi) = \mathbb{E}_{\xi\sim\pi}\left[\min_{i\in[n]} f_\xi(\theta_i)\right]$. □

*Proof of Theorem 2.3.* Note that

$$W_f(\hat{\rho}, \pi) - L^* = \inf_{\mu\in\Pi(\hat{\rho},\pi)} \mathbb{E}_{(\theta,\xi)\sim\mu}[(f_\xi(\theta_i) - f_\xi(\theta_\xi))].$$

The result then follows immediately from Assumption 2.2 and the definition of $p$-Wasserstein distance. Therefore, for any $\boldsymbol{\theta}$ and $\boldsymbol{\nu}$,

$$W_{p_1}(\hat{\rho}_{\boldsymbol{\theta}^*,\boldsymbol{\nu}^*}, \rho^*) \leq \frac{1}{h_1}(W_f(\hat{\rho}_{\boldsymbol{\theta}^*,\boldsymbol{\nu}^*}, \pi) - L^*) \leq \frac{1}{h_1}(L(\hat{\rho}_{\boldsymbol{\theta},\boldsymbol{\nu}}, \pi) - L^*) \leq \frac{h_2}{h_1}W_{p_2}(\hat{\rho}_{\boldsymbol{\theta},\boldsymbol{\nu}}, \rho^*),$$

which yields (5). □