# OpenReview forum: "argmax centroid"
_NeurIPS.cc/2021/Conference — NeurIPS 2021 Poster_

### Official Review · Reviewer_ZWfd · 2021-07-16

**Rating:** 7
**Confidence:** 4

**Summary:**

The paper proposes a generic method for approximating the argmax distribution of a random function using optimized centroids without explicitly sampling from the argmax distribution. While the closest application for the method is in meta-learning and multi-task learning, the proposed method can find application as an alternative to bootstrap among other applications. In short, for a random function $f_\xi(\theta)$ (think of $\xi$ as data, $\theta$ as parameters, and $f_\xi(\theta)$ as the loss function), the method finds a set of parameters $\boldsymbol{\theta}:=\\{ \theta_i\\}_{i=1}^n$
such that it minimizes
$E_\{\xi\sim\pi\}\[ min_\{i\in\\{1,...,n\\}\} f_\xi(\theta_i)\]$. During inference and for a new task, the authors first find the appropriate parameters for the task (based on a validation set) from the set $\boldsymbol{\theta}$ and then use the corresponding model to solve the task. The method is shown to minimize a bound in Wasserstein distance between the approximated argmax distribution and the real distribution. Finally, the authors show the method's superior performance in various numerical experiments spanning few-shot supervised learning, personalized dialogue systems, and multi-target domain adaptation compared to various baselines.

**Limitations And Societal Impact:**

The authors state that "Our algorithm slightly increases the time cost and memory cost in the experiments," however no rigorous computational analysis is provided in the paper.

**Main Review:**

**Strengths**

* The paper provides a unifying and interesting perspective for various multi-task learning applications by studying the argmax distribution of a random function
* The method can be interpreted as an implicit clustering for tasks, where similar tasks are assigned to the same model parameters
* By assigning different parameters to different 'clusters of tasks,' the proposed method addresses one of the main shortcomings of meta-learning approaches (like MAML), which assume similarity/homogeneity of training tasks.
* The authors provide extensive numerical experiments to demonstrate the practicality of the proposed approach.

**Weaknesses**

* argmin and argmax are used interchangeably. While this is not a major issue, picking one and sticking with it throughout the paper will improve consistency.
* The paper does not report any insights regarding the computational cost of the approach, specifically in comparison with meta-learning approaches like MAML and Reptile (which is more efficient). I think it might be safe to say that for $n=1$ the method is technically a classic meta-learning approach, but does the computation linearly scales in $n$?
* The model selection required for solving new tasks seems to be constraining. When the argmax distribution is approximated on supervised learning tasks, but we don't have any training/val/test labels for the new tasks, how can one find the right model for the new task?

**Summary**

The paper provides an interesting perspective on estimating the argmax distribution of a random function, which finds application in many machine learning problems involving learning multiple tasks/datasets. The paper is well written (minus a few typos here and there) and is interesting. The authors show that the proposed approach theoretically minimizes a Wasserstein bound between their approximation and the true distribution of argmax.  Moreover, the method is validated on various multi-task learning problems. While the paper is in good shape, it lacks in certain aspects. For instance, it doesn't give a sense of the computational scalability of the algorithm. Also, the task clustering aspect is exciting. The authors could have provided the tasks that share a model to give a sense to the reader about the implicit modeling of task similarities. I think the paper is interesting to the broad ML community, and despite its mentioned deficiencies, it stands above the acceptance threshold.

**Minor editorial comments**
Please proofread the paper and address the typos. Here are some that I caught:
* line 242: "in and ensemble" should be "in an ensemble"
* line 356: "increase" should be "increases"

**Time Spent Reviewing:**

3 hrs

---

> ### Author Response · Authors · 2021-08-10
> **Reply to Reviewer ZWfd**
>
> We thank the reviewer for the valuable feedback and for pointing out the typos.
>
> 1.argmin and argmax are used interchangeably. While this is not a major issue, picking one and sticking with it throughout the paper will improve consistency.
>
> Thanks for your suggestion. We will make it consistent in revision.
>
> 2. The paper does not report any insights regarding the computational cost of the approach, specifically in comparison with meta-learning approaches like MAML and Reptile (which is more efficient). I think it might be safe to say that for n=1 the method is technically a classic meta-learning approach, but does the computation linearly scales in n?
>
> Thanks. We will add a detailed discussion on computational cost. As the steps of Algorithm 1 imply, the complexity per step is $A * n + B$, where $A$ is the cost of forward propagation (for picking the best model in Line 3), and $B$ is the cost of backward propagation (for updating $\theta$ in Line 4), and $A$ is smaller than $B$ in practice. We think we are similar to other off-the-shelf ensemble methods in terms of complexity.
>
> 3. The model selection required for solving new tasks seems to be constraining. When the argmax distribution is approximated on supervised learning tasks, but we don't have any training/val/test labels for the new tasks, how can one find the right model for the new task?
>
> Thanks for your suggestions. If the new task is fully unlabeled, we think it is natural to pick the best model using the (set of) old (labeled) tasks that are most similar to the new task under a proper similarity measure. This is an interesting direction and we will explore this issue further.

---

> > ### Comment · Reviewer_ZWfd · 2021-08-19
> > **Post rebuttal comments**
> >
> > I thank the authors for providing a rebuttal. Besides the point that the computation linearly scales with $n$, the rebuttal doesn't change my original assessment of the paper. Therefore, I stand with my previous score.

---

> > > ### Author Response · Authors · 2021-09-03
> > > **Thanks**
> > >
> > > Dear Reviewer ZWfd, we want to thank you again for your comments and efforts. We will provide a detailed discussion on computation cost, as well as the case of no labels in new tasks in the revision.

---

### Official Review · Reviewer_a626 · 2021-07-17

**Rating:** 6
**Confidence:** 4

**Summary:**

A method is presented in the paper to approximate the distribution of argmax of functions depending on additional random variables. The idea is to use a set to represent centroids to approximate the argmax distribution, and in training iterations and test phase assign each realization one member of the set.

**Limitations And Societal Impact:**

yes

**Main Review:**

+:

The proposed method is interesting, and a theoretical interpretation is provided in the paper. The experiments on setups like few-shot image classification and multi-target domain adaptation show that integrating the proposed procedure with existing methods can improve their performance. Additionally, an ablation study is done comparing the method with some ensemble methods showing reasonable improvement.


-:

The first part of the paper leads the reader to expect that with higher n better representation of the argmax distribution and better results can be achieved, while the results are not consistent in that regard. In my opinion, the authors need to further discuss this in the paper.

The experiments need to be strengthened by an ablation study to show the effect of selecting a model at test phase vs. the training procedure to form the models, for example, by picking best model for other ensemble approaches.

While the current experiments are valuable, in my view it is a bit unfortunate that the focus is only on multi-task/domain learning. The motivation and initial discussions about the method are very general, for example, bootstrap and Thompson sampling are mentioned several times to motivate the approach, but no experiment is performed on them. The authors suggest the method can be applicable to a single task setup, so an evaluation of it can be added.

The experimental details do not seem enough to easily reproduce the results and some details like training/run time and compute resources are not provided. The authors can provide more details in the supplementary.

-------------------------------
After the rebuttal, I still think the ablation study and additional experiments will be helpful in supporting the claims. After looking through the paper again and assuming the authors will incorporate reviewers’ suggestions, I think the paper can be useful to the community, especially its way of considering the meta-learning and multi-task settings. Hence, I have increased the score.


**Time Spent Reviewing:**

2.25

---

> ### Author Response · Authors · 2021-08-10
> **Reply to Reviewer a626**
>
> Thanks for your comments.
>
> 1. The first part of the paper leads the reader to expect that with higher n better representation of the argmax distribution and better results can be achieved, while the results are not consistent in that regard. In my opinion, the authors need to further discuss this in the paper.
>
> We will clarify this issue in the revision. Larger n will yield better solution of the argmax approximation (which can be roughly viewed as the training objective), but the practical performance, which is the testing result, do not necessarily increase with n, because of overfitting, domain shift and other reasons. This is a common phenomenon and applies equally to standard ensemble methods as well.
>
>
> 2. While the current experiments are valuable, in my view it is a bit unfortunate that the focus is only on multi-task/domain learning. The motivation and initial discussions about the method are very general, for example, bootstrap and Thompson sampling are mentioned several times to motivate the approach, but no experiment is performed on them. The authors suggest the method can be applicable to a single task setup, so an evaluation of it can be added.
>
> Thanks for your advice. We will add more discussions and comparisons about these approaches and in the experiments. Currently, we mainly focus on deep learning related topics and therefore do not include empirical comparison with these methods. We hope the reviewer understand that we have already included a quite large set of experiments and that it is not possible to enumerate every all possible aspects in the first paper that introduce the basic methodology. We do appreciate your comments and improve the draft to avoid potential confusions.
>
> 3. The experimental details do not seem enough to easily reproduce the results and some details like training/run time and compute resources are not provided. The authors can provide more details in the supplementary.
>
> Thanks for your suggestions. We will release the code in revision. And we will improve the appendix with more detail information.

---

### Official Review · Reviewer_kFgf · 2021-07-17

**Rating:** 5
**Confidence:** 2

**Summary:**




This paper proposes optimizing centroid points to compactly approximate the argmax distribution with a simple objective function. The proposed method is theoretically minimizing a bound of Wasserstein distance between the empirical distribution of the centroids and the ground-truth distribution. Argmax centroids have many applications for machine learning tasks; the author validates the proposed method on multiple meta-learning and mult-task learning tasks.



**Limitations And Societal Impact:**

-

**Main Review:**

My main issue is with the empirical evaluation. I am not an expert in centroid approximation, so I leave that to other reviewers to judge the originalities and soundness of the proposed method.

- For the personalized dialog experiments, the results are not consistent with the original paper. Which metric is case-insensitive? Only BLEU (as described in line 279) or both (described in the title of table 4)?

- Since multi-task learning is a natural application of argmax centroid, it will be interesting to extend the experiments with multi-task learning on SuperGLUE using pre-traineds model to test the adaptability of the proposed framework.

- Some aspects of the paper's presentation can be improved. E.g., Line 343, incorrect ref. "discriminator loss in (??)", confusing references on table 2 and table 3 in section 4.2.

**Time Spent Reviewing:**

12

---

> ### Author Response · Authors · 2021-08-10
> **Reply to Reviewer kFgf**
>
> Thanks for your comments.
>
>  1. For the personalized dialog experiments, the results are not consistent with the original paper. Which metric is case-insensitive? Only BLEU (as described in line 279) or both (described in the title of table 4)?
>
> As we mentioned in the paper, we use the BERT pretrained model and the original paper does not. All these metrics are case-insensitive. We will add the detailed description on this.
>
>  2. Since multi-task learning is a natural application of argmax centroid, it will be interesting to extend the experiments with multi-task learning on SuperGLUE using pre-traineds model to test the adaptability of the proposed framework.
>
> Thanks for your suggestion. We agree that applying our method on SuperGLUE is interesting. We plan to explore more NLP applications in future work.
>
>
>  3. Some aspects of the paper's presentation can be improved. E.g., Line 343, incorrect ref. "discriminator loss in (??)", confusing references on table 2 and table 3 in section 4.2.
>
> We will clarify these typos.

---

> > ### Comment · Reviewer_kFgf · 2021-08-29
> > **Post rebuttal comments**
> >
> > Thanks for the authors' response. It is preferred to have the same experimental setting as the baseline paper. So I keep the score unchanged.

---

### Official Review · Reviewer_pFrh · 2021-07-18

**Rating:** 5
**Confidence:** 4

**Summary:**

The paper proposed a method that optimizes a set of centroid points that approximates the argmax distribution p*. Rather than using the Monte Carlo sampling that draws samples randomly, their approach choose the location of each points that approximate target distribution p*.  This approach can be an alternative for bootstrap and can be applied to deep learning applications. The paper showed the effectiveness of the proposed method on few shot image classification, personalized dialogue systems and multi-target domain adaptation. The proposed algorithm boosts the SOTA performance for few-shot classification and meta learning tasks.



**Limitations And Societal Impact:**

1-  The paper proposed a centroid-like approximation to replace Monte Carlo sampling. At each iteration, the algorithm draws a set of random variables, finds \theta_i, and updates \theta_i with gradient descent. It also keep track of frequency which later on used for the estimation of distribution p*. This approach is more sample efficient than the simple Monte Carlo sampling but each step is expensive. My concern is that it may need less samples but eventually takes same time to estimate the p*. To have a better comparison, I suggest not only showing how Wasserstein distance is changing by increasing the number of samples but also the time and memory cost.

2- The proposed algorithm uses the best single model selected from the pool during evaluation. Specifically, It trains a fixed number of classification heads (the argmax centroids) and it picks the best head and further finetune it on the training set for each test task during testing.It may lead to a high computation cost in comparison to existing methods. Adding some results/datapoints should make it clear if this approach is practical.

3- Existing ensemble learning averages the output of multiple models, while this approach picks the best single model. I am wondering if relaxing the best to average would reduce the computation cost!

4- As discussed in the paper, if some \theta_i is rarely selected, it's important weight gets close to zero. The paper addressed this problem with resampling a set of new \theta_i with replacement and repeating the updates. But if the total samples are small, this approach never detects those rare samples and may end up with a very biased estimation of the argmax distribution. In experimental results some of these details are missing. If the sample size is very large then the simple Monte Carlo may also work!


**Main Review:**

1- The paper is well written and it well motivates the application of approximating the argmax distribution in different deep learning tasks.

2- Proposed approach optimizes a set of centroid points that approximates the argmax distribution rather than random sampling, and with a toy example, it shows that their approach yields lower Wasserstein distance than random sampling.

3- The effectiveness of their method is evaluated on multi-task learning applications, including few shot image classification, personalized dialogue systems and multi-target domain adaptation. Their result shows that the proposed algorithm improves the baseline methods in different settings.


**Time Spent Reviewing:**

6 hours

---

> ### Author Response · Authors · 2021-08-10
> **Reply to Reviewer pFrh**
>
> We thank the reviewer for the valuable feedback and pointing out the typos. We will improve the draft based on your comments.
>
> 1. “To have a better comparison, I suggest not only showing how Wasserstein distance is changing by increasing the number of samples but also the time and memory cost.”
>
> Our method is simple and fast. As the steps of Algorithm 1 imply, the complexity per step is $A * n + B$, where $A$ is the cost of forward propagation (for picking the best model in Line 3), and $B$ is the cost of backward propagation (for updating $\theta$ in Line 4), and $A$ is smaller than $B$ in practice. We think we are similar to other off-the-shelf ensemble methods in terms of complexity.
>
> We mention that the naive method of calculating $n$ independent optimization problems could be much more expensive, which is exactly we want to avoid in this work.
>
> 2. “The proposed algorithm uses the best single model selected from the pool during evaluation. Specifically, It trains a fixed number of classification heads (the argmax centroids) and it picks the best head and further finetune it on the training set for each test task during testing. It may lead to a high computation cost in comparison to existing methods. Adding some results/datapoints should make it clear if this approach is practical.”
>
> Thanks for your advice. Current few-shot learning methods use a fixed backbone and a lightweight head. Therefore, training a lot of heads would not increase the training cost by a large margin. For our few-shot image classification cases, it only increases the training cost by around 1% ~ 5% for different methods and different numbers of heads.
>
> 3. “Existing ensemble learning averages the output of multiple models, while this approach picks the best single model. I am wondering if relaxing the best to average would reduce the computation cost!”
>
> Sorry, we do not get your point here. During training, both approaches will train all the models. During the evaluation, picking the best model reduces the time cost of averaging all the models.
>
> 4. “As discussed in the paper, if some \theta_i is rarely selected, it's important weight gets close to zero. The paper addressed this problem with resampling a set of new \theta_i with replacement and repeating the updates. But if the total samples are small, this approach never detects those rare samples and may end up with a very biased estimation of the argmax distribution. In experimental results some of these details are missing. If the sample size is very large then the simple Monte Carlo may also work!”
>
> Thanks for your suggestions. On the toy cases (section 4.1), we show that for a small number of samples, we can improve the MC sampling. For a very large sample size, e.g. 10000, these two approaches have similar performance. We would add additional results and these discussions in revision.

---

> ### Author Response · Authors · 2021-09-03
> **Reply to Reviewer pFrh**
>
> Dear reviewer pFrh, thank you again for your attention to our work. We were wondering if our response addressed your concerns properly.
> We would be very grateful if you could let me know what you think and if any issues remain.
>
> It seems that your questions were mainly about the issue of computational cost. Regarding these, the main points in our response were:
>
> 1)  In terms of solving the random argmax problem, our method is (designed to be) much be computationally efficient than the naive algorithm of repeatedly (and separately) optimizing i.i.d. drawn random functions.
>
> 2) In terms of learning ensemble neural networks, our method is at least as efficient as the popular ensemble methods;  In fact, it can be more efficient than the standard averaging-based ensemble methods because 1) we only need to backpropage the best centroid for each data point during training, and 2) we pick the best model (rather than averaging all the models) and is hence faster during the inference time.
>
> Thank you very much for your valuable feedback and efforts.

---

### Official Review · Reviewer_TLpj · 2021-07-24

**Rating:** 6
**Confidence:** 3

**Summary:**

This paper proposes a method called Argmax Centroids for learning an approximation to the distribution of the minimizer of a random function. The central idea is to learn a set a particles such that the expected distance between the minimizer of a randomly drawn function and the closest particle is small. The method is shown to be connected theoretically to Wasserstein distance. Experiments on few-shot classification, personalized dialogue modeling, and multi-target domain adaptation demonstrate improvements when the technique is applied to baseline models.

**Ethical Concerns:**

No ethical concerns.

**Limitations And Societal Impact:**

The authors do not touch upon negative societal impact. However, the method is fairly general and therefore it seems quite difficult to predict what negative impacts there will be.

**Main Review:**

The formulation of the problem (learning the distribution of a random function's minimizer) encompasses a wide range of applications and should be of broad interest. The paper is overall clear and well-written.

The experiments were thoughtfully chosen to illustrate the appeal of the method. It can be applied in a straightforward manner to various different choices of model and random functions. The authors' choice to view meta-learning and multi-target domain adaptation from a random function perspective was insightful. Unlike the dialogue personalization and domain adaptation experiments, the few-shot learning results did not show a large win for Argmax Centroids. However, the fact that Argmax Centroids is versatile enough to be applied to various models is a point in its favor.

Despite the aforementioned strengths, there are some weaknesses of this paper. Firstly, the related work does not connect Argmax Centroids well to prior work. In particular, the connection to Perturb-and-MAP models (which also consider optimization of random functions) and particle filters (which also perform inference with parameter samples) could be expanded upon.

I also have concerns about some details in §2. The first is the upper bound in the proof of Theorem 2.2 (2nd line of the proof). For the lower bound, the min is moved outside of $|| \theta_i - \theta_\xi ||^{p_1}$, which makes sense. But moving the min outside in the same way for $|| \theta_i - \theta_\xi ||^{p_2}$ possibly makes the bound tighter and thus does not follow from Assumption 2.1. Similarly, in the 3rd line of the proof, the order of the expectation and the min is changed. In general, the expectation of a minimum will not be the same as the minimum of an expectation. Finally, is there a citation for the expression for the optimal $\nu$ in the fourth line of the proof? As an aside, the definition of Wasserstein should probably be stated somewhere in §2.

ll. 256-258: The need for selecting the best model on the training set does not make much sense to me. Isn't the $\theta_i$ selected based on the support set for the current episode?

Overall, the aim of the paper is well-motivated and the method is simple enough to implement yet has the potential for broad impact. However, my concerns about related work and the proof of Theorem 2.2 lead me to not recommend acceptance in the paper's current state. I hope these points will be addressed during the discussion phase.

=====
After reading the authors' comments, my concerns about Theorem 2.2 have been addressed, assuming the authors include a derivation of lines 3-4 of the proof in the appendix.

**Time Spent Reviewing:**

3.5

---

> ### Author Response · Authors · 2021-08-10
> **Reply to  Reviewer TLpj**
>
> Thanks for your comments.
>
> **Regarding Perturb-and-MAP** We have cited the perturb-and-MAP (or random MAP) works and will include more citations and draw more discussions in the revision. I think it perturb-and-MAP is certainly related to our idea in high concept, but the setting is very different because perturb-and-MAP is very much an approximate inference and learning framework for graphical models, but our method focuses on addressing meta learning problems.
>
> **Proof of Theorem 2.2**
> Our proof is correct. There is no concern of exchanging orders of min and expectation in the two places you worried about.
>
> The second line of the proof holds because $\Phi(g):= E_{\xi}[\min_i g(\theta_i, \theta_\xi)]$ is a monotonic operator of function $g$, that is, if $g_2(\theta_i, \theta_\xi) \leq g(\theta_i, \theta_\xi)$ for any $(\theta_i,\theta_\xi)$, we have $\Phi(g_1) \leq \Phi(g_2)$. The result then follows by taking $g(\theta_i, \theta_\xi) = f(\theta_i) - f(\theta_\xi)$ and using Assumption 2.1.
>
> The fact that $E[\min_i ||{ \theta_i - \theta_{\xi}}||^p ]  =  \min_{\nu \in \mathcal V} W_p(\hat \rho_{\theta, \nu}, ~~  \rho^*)^p$ in the third line of the proof is correct --- it is a special property of Wasserstein distance that is related to the fact that K-means minimizes the optimally weighed Wasserstein distance (because the left hand side of the equation is a K-means loss function, say, when $p=2$).  This is a folklore result and we will include the full proof of it in the Appendix. Basically, the idea of the proof is that the transport map between $\hat{\rho}_{\theta, \nu}$ and $ {\rho}^*$, when the weights $\nu$ are optimized, is to assign each point drawn from $\rho^*$ to the closest point $\theta_i$, and hence correspondingly, the optimal choice of $\nu$ is $\nu_i =E[I \in \arg\min_i ||\theta_i - \theta||^p]$ as we stated in the proof.
>
> **Selecting the best model**
> What we Describe in Line 256-258 is about how to select the best model to use **during the testing phase**, which is different from the training procedure that we described earlier. So after the training, we obtain a set of argmax centroids $\{\theta_i\}$. Then, during the testing phase, when we are given a new task (with a separate training and testing dataset), we need to decide which model (i.e., $\theta_i$) we will pick to use (and note our method is different from typical ensemble learning which uses the average of the outputs of all $\theta_i$).   We choose the best $\theta_i$ based on the training set (which could be small) of the new task.
>
> We realized that we made the proof too brief (due to the space limit). We will write a more detailed proof in the revision. But our technical results are correct.

---

> > ### Comment · Reviewer_TLpj · 2021-08-23
> > **Response to reviewers**
> >
> > Thanks to the authors for their response. After reading the authors' comments, my original statement regarding the min of the upper bound was incorrect and I now agree with the authors on this point. Regarding the folklore result, I understand the high level intuition but a proof would be appreciated. The point about selecting the best model has also been sufficiently clarified in my mind. I am willing to revise my score upwards, assuming the folklore result will be included in the appendix.

---

> > > ### Author Response · Authors · 2021-09-03
> > > **Thanks! and More about K-means & Wasserstein**
> > >
> > > Dear Reviewer, thank you a lot for considering our response. We will definitely include detailed proofs of the folklore result on K-means and Wasserstein, as well as the other results in the paper. It is on our best interest to make our the proofs as clean and as self-contained as possible and we appreciate that the reviewer raised the points.
> > >
> > > Regarding the connection on K-means and Wasserstein, here is some quick intuition (the full proof requires to set up more notation and we will defer it to the appendix). Assume $\{\theta_k\}$ is a set of centers, and $\{x_i\}$ is a set of data points (in our setting these are the random $\theta_\xi$). Then the optimal transport schedule between the centers and the data points is to simply transport each data point $x_i$ to its closest center $\theta_{k_i^*}$, where $k_i^* = \arg\min_{k} ||\theta_k - x_i||$. In this way, the percentage of data points that are transported to $\theta_\ell$ equals $v_\ell = E_{x\sim Data}[ \ell= \arg\min_{k} ||\theta_k - x||]$.
> > >
> > > Then, one can show that this optimal transport procedure can be exactly interpreted as the solution of the optimally weighted Wasserstein distance, following the definition of Wasserstein metrics.

---

### Official Review · Reviewer_Zrxe · 2021-07-26

**Rating:** 6
**Confidence:** 4

**Summary:**

This paper proposes a natural and simple (yet general) method for approximating the argmax distribution through optimizing a set of centroid points. It is also shown theoretically that under suitable assumptions such an approximation minimizes a bound based on the Wasserstein distance with the actual argmax distribution. Extensive experimentation is carried to support the usefulness of the method. The data include a toy dataset, a few shot learning task, a dialogue system task and a domain adaptation task.

**Limitations And Societal Impact:**

Limitations are discussed.

**Main Review:**

The central object of interest in this paper is the argmax distribution, which appears frequently in different contexts in machine learning. More specifically, given a random function f_\xi(\theta) with \xi representing a random variable and \theta the variables of interest, we want to estimate a distribution \rho^\ast of the optimal points when \xi is drawn randomly. This problem is approached in quite a natural manner: By seeking a set of centroid points from \theta (optimized for explicitly) such that they are close to \rho^\ast. Such an approach permits avoiding expensive Monte Carlo sampling based methods. The MC approximations are also poor when the number of sampled points are small.

The solution to the above problem is quite simple and straightforward. The objective function and method is summarized in equations (2) and (3). We optimize for the \theta (which are also assigned importance weight so as to approximate \rho^\ast. As might be expected, the objective is quite similar to the k-means objective, except that the distortion function is for distributions, and in particular for the argmax distribution.  The whole procedure is summarized in algorithm 1. Further, it is shown theoretically that the method can be seen as minimizing a bound on the Wasserstein distance between the empirical measure of the centroids and the true distribution (rho^\ast above). Given the usage of importance weights, and a simple distortion based objective, this is simple to see.

Then a set of experiments are presented that show that the above procedure indeed leads to gains. The first experiment is a toy dataset composed of a mixture of gaussians. It is shown that the method yields a smaller Wasserstein distance with the true distribution with few number of samples as compared to a monte carlo approximation. Then is a few-shot task akin to that in the MAML (and following) papers. Using some recently proposed methods as the baseline, the centroid approximation is shown to have consistent performance improvements. Similar results are shown for a standard personalized dialogue systems task and a domain adaptation task.

In summary the paper proposes a very simple and natural method for approximating the argmax distribution. The theoretical contributions are also straightforward. From a purely technical perspective (in the sense of the tools used, originality of the method etc), the contributions are also straightforward. From my perspective the main contribution of the paper is experimental -- showing that such a simple method will lead to performance gains in a variety of tasks, which I believe is sufficiently corroborated.


Minor comments:
- Line 48: "theoretically and show its property." --> this sentence is a little bit awkward, please reword this.
- Line 68: "important weight" --> importance weight.
- Line 77: "nearly zero important weight" --> same as above

**Time Spent Reviewing:**

3 hours

---

> ### Author Response · Authors · 2021-08-10
> **Authors' Response to Reviewer Zrxe**
>
> Thanks a lot for your advice. We would correct these typos and highlight our contribution in the introduction part.

---

### Decision · Program_Chairs · 2021-09-27

**Decision:**

Accept (Poster)

**Comment:**

This paper is well motivated by the problem of learning the distribution of a random function's minimizer)  which has a wide range of applications. The paper made a solid contribution to this important problem by proposing an interesting and novel method (argmax centroids) with theoretical guarantees and experimental validations. During the rebuttal, concerns regarding the proofs of theorem 2.2 were addressed.
The final version should address the following points as well as other suggestions made by the reviewers:

1. correcting these typos and highlighting the contribution in the introduction part.
2. Discussion of the limitation of choosing the best model using a labeled set of test data
3. providing detailed proofs of the folklore result on K-means and Wasserstein